# A systematic review and meta-analysis of unimodal and multimodal predation risk assessment in birds

Kimberley J. Mathot [1,2] ✉, Josue David Arteaga-Torres [1], Anne Besson[1,5], Deborah M. Hawkshaw [1], Natasha Klappstein[1,6], Rebekah A. McKinnon[1], Sheeraja Sridharan[1] & Shinichi Nakagawa [3,4]

Despite a wealth of studies documenting prey responses to perceived predation risk, researchers have only recently begun to consider how prey integrate information from multiple cues in their assessment of risk. We conduct a systematic review and meta-analysis of studies that experimentally manipulated perceived predation risk in birds and evaluate support for three alternative models of cue integration: redundancy/equivalence, enhancement, and antagonism. One key insight from our analysis is that the current theory, generally applied to study cue integration in animals, is incomplete. These theories specify the effects of increasing information level on mean, but not variance, in responses. In contrast, we show that providing multiple complementary cues of predation risk simultaneously does not affect mean response. Instead, as information richness increases, populations appear to assess risk more accurately, resulting in lower among-population variance in response to manipulations of perceived predation risk. We show that this may arise via a statistical process called maximum-likelihood estimation (MLE) integration. Our meta-analysis illustrates how explicit consideration of variance in responses can yield important biological insights.

A wealth of research demonstrates that prey can use a range of cue types when assessing predation risk[1]. Different cues can indicate different average levels of risk, and numerous studies have shown that animals respond more strongly to cues indicating higher average risk. For example, animals have a stronger response to more lethal predator types[2–5] and also respond more strongly to predators exhibiting behaviours/postures that indicate more imminent risk[6–8]. For example, damselfish (*Stegastes planifrons*) exhibit stronger avoidance responses to predator models that are oriented in a strike pose (indicating high immediate risk) compared to predator models that are oriented in non-attacking postures (indicating lower immediate risk)[9].

Two cues may indicate the same average risk level, but differ in the certainty that they confer to prey. Theory predicts that the information quality of a cue (i.e., certainty about current predation risk conferred by a given cue) should also affect the magnitude of response[10–13]. In some cases, uncertainty about predation risk may lead to behavioural over-responses (i.e., overestimation of risk), because of the asymmetrical cost of overresponse (i.e., missed foraging opportunity), versus cost of under-response (i.e., injury or death)[14], favours overresponse. However, in other cases, full responses to uncertain predation risk may be relatively costly, such as if foragers will experience a high risk of starvation if feeding is unnecessarily interrupted[12].

[1]Department of Biological Sciences, University of Alberta, Edmonton, AB, Canada. [2]Canada Research Chair in Integrative Ecology, Department of Biological Sciences, University of Alberta, Edmonton, AB, Canada. [3]Evolution & Ecology Research Centre and School of Biological, Earth and Environmental Sciences, University of New South Wales, Sydney, NSW 2052, Australia. [4]Theoretical Sciences Visiting Program, Okinawa Institute of Science and Technology Graduate University, Okinawa, Onna 904-0495, Japan. [5]Present address: Department of Zoology, University of Otago, Otago, New Zealand. [6]Present address: Department of Statistics, Dalhousie University, Halifax, NS, Canada. ✉e-mail: mathot@ualberta.ca

In such cases, cues indicating a given level of predation risk with high certainty should elicit stronger responses compared to cues indicating the same level of predation risk with lower certainty[10–12], though empirical tests of this prediction are lacking[12].

More recently, researchers have begun to address how prey integrate information from multiple cues in their assessment of risk[10,12,15,16]. Patterns of multimodal cue integration can broadly be grouped into three types of integration: redundancy/equivalence, enhancement and antagonism[12]. The expected outcome of multimodal cue integration depends on the level of uncertainty associated with each cue on its own relative to the uncertainty that results from the combined cues. Equivalence (or redundancy) describes the scenario in which the response elicited by either cue on its own is the same as the response elicited by two cues combined[12]. If the unimodal cues differ in the response they elicit, for example, because one provides greater certainty about current risk, then we would expect the two cues combined to elicit the same response as the higher certainty unimodal cue, in which case, the response might be described as 'redundant' (Fig. 1A). Equivalence (or redundancy) is expected when the addition of a second cue provides no greater certainty about the current level of threat than the high certainty cue on its own, nor does it change the estimated risk[12]. Alternatively, combined cues may result in enhancement, whereby the response to the combined cues is greater than the response to either cue on its own (Fig. 1B). This is expected when two cues together indicate a higher likelihood than either cue on its own[12]. Finally, multiple cues can combine to produce antagonistic effects, whereby the response to the combined cues is less than the response to the higher certainty cue on its own or even lower than both cues (Fig. 1C). This is expected to occur when the combination of cues increases the certainty that predation risk is low relative to either cue on their own[12].

Here, we conduct a systematic review and meta-analysis of studies that experimentally manipulate perceived predation risk in birds with unimodal or multimodal cues of predation to test predictions derived from the uncertainty reduction framework described above. We restrict our review to birds because their anti-predator responses have been studied extensively, providing a large number of studies with relatively comparable experimental designs. We use these studies to evaluate support for two predictions from the uncertainty reduction framework. First, we test the prediction that anti-predator responses to unimodal cues of current predation risk increase with increasing cue certainty[10–12,15]. The three most common cue modalities used in experimental manipulations of perceived risk in avian studies are visual (e.g., predator mounts), acoustic (e.g., mobbing calls, predator calls) and chemical (e.g., natural or synthetic olfactory predator cues). Visual cues of predation provide high certainty information that the predator is present and may also provide postural or behavioural cues as to the predator's current state[6–8]. Mobbing calls are uncertain

because they can be given as false alarms[17,18]. However, when produced honestly, mobbing calls may convey information about the level of threat the predator poses[5] and also convey that the threat is currently being attended to[15]. Acoustic cues made by predators themselves (e.g., calls) provide information that a predator is in the area. However, because predators generally do not vocalise when hunting (e.g., ref. [15]), these cues are generally used to evaluate longer-term responses to increased perceived predator abundance, for which they provide low certainty information about future predation risk. Predator chemical cues may convey information about predator type, and predator diet and importantly, chemical cues are not limited by visual obstructions[19–21]. In birds, chemical cues may be detected directly from the predator, but may also be detected indirectly, such from faeces left in the area, or transfer of chemicals to surfaces which birds come into contact with (e.g., nestboxes, feeders). In the latter case, an olfactory predator cue provides information that a predator has been present in an area, but not whether it is currently in the area or if it is, in what state (e.g., hungry or sated). As the latter cues are the type used in experimental manipulations of perceived predation risk in birds, we assumed that visual cues (e.g., predator mounts) provide greater certainty compared to acoustic cues (e.g., mobbing calls), which provide greater certainty compared to chemical cues (e.g., predator odour). Following this assumption, for unimodal cues, we predicted that antipredator responses would be greatest in response to visual cues, intermediate in response to acoustic cues and lowest in response to chemical cues, based on the assumption that over-response to low certainty risk would be costly[12].

Second, we evaluated support for specific forms of cue integration. We predicted redundancy between visual and chemical cues because adding a chemical cue to a visual cue indicating that a predator is currently present should not provide any further reduction in uncertainty regarding current predation risk compared to the direct observation of a predator alone (Fig. 1A). We predicted enhancement between acoustic cues and chemical cues. On their own, acoustic and chemical cues, each provides uncertain information about whether a predator is present. Thus, receiving both cues simultaneously should increase the certainty that a predator is currently present, resulting in an elevated response (Fig. 1B). Finally, we predicted antagonistic integration between visual and acoustic cues. On their own, visual cues provide greater certainty that a predator is currently present compared to mobbing calls for the reasons outlined above. However, acoustic cues such as mobbing calls presented in combination with visual cues could lower perceived risk compared to the visual cue alone by providing information that the threat is already being attended to, by increasing real or perceived group size and thereby providing dilution of risk[22], or both (Fig. 1C). Our meta-analysis shows that providing multiple complementary cues of predation risk simultaneously does not

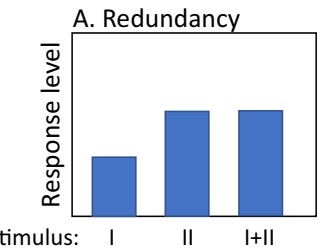
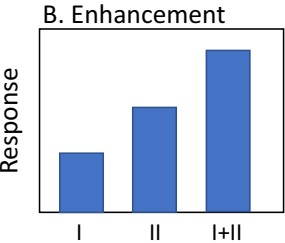
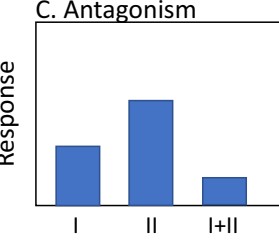

**Fig. 1 | Illustration of three types of multimodal cue integration.** We assume that the unimodal cues differ in information quality (i.e., certainty), such that stimulus II has higher certainty and elicits a stronger response on its own compared to stimulus I. **A** This illustrates signal redundancy (or equivalence), whereby the multimodal stimulus does not increase certainty relative to the higher certainty stimulus (II) on its own. **B** This illustrates enhancement, where the multimodal stimulus increases certainty relative to either stimulus on their own, thereby eliciting a stronger response. **C** This illustrates antagonism, whereby the multimodal cue results in a lower estimation of risk than the more certain unimodal cue on its own. Note that any reduction in the response to the multimodal cue relative to the more certain stimulus (II) would be considered antagonism even if it is higher than the response to the lower certainty cue (I).

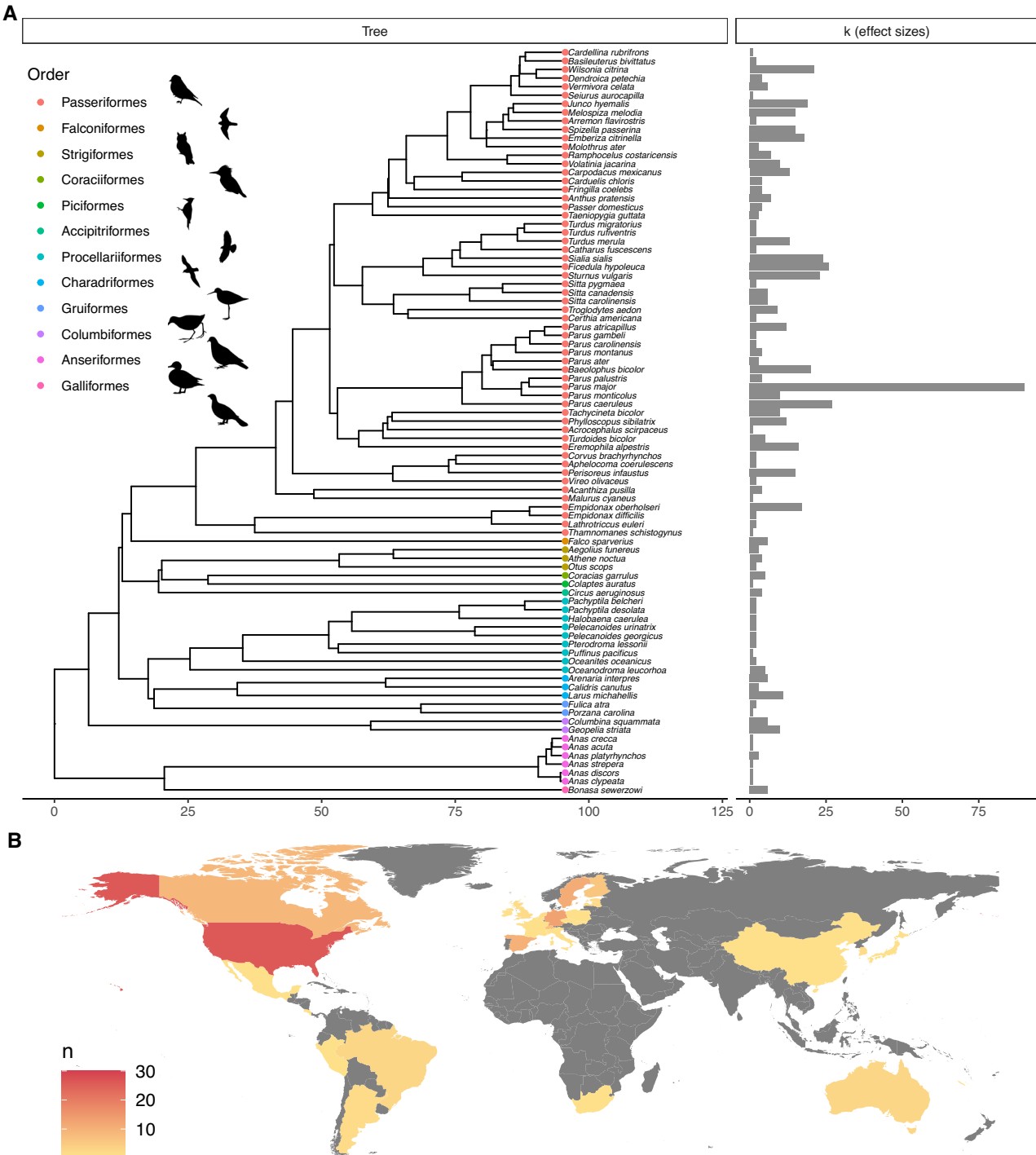

**Fig. 2 | Illustration of phylogenetic and geographic breadth of estimates included in meta-analysis. A** Shows the phylogenetic relationships used in the meta-regression, grouped by order and the associated mean effect size for response to manipulations of perceived predation risk for k estimates from K studies. **B** Shows the geographic distribution of studies, where the colour of the country on a gradient from yellow to red represents the total number of studies (*n*). Grey is used for countries from which no estimates were obtained. Silhouettes representing different bird orders were obtained from PhyloPics, with artist credits and copyright: T. Michale Keesey (PDM 1.0), Ferran Sayol (CC0 1.0), Gabriela Palomo-Munoz (CC BY-NC 3.0) Andy Wilson (CC0 1.0) and Alexandre Vong (CC0 1.0). Detailed copyright information for all images can be accessed at: https://www.phylopic.org/permalinks/4d2aebec1e2f2da818396c344eb377c61d6ce0d70ddb15d09d7671defdf00ed2.

affect mean response, but does reduce variance in response. In other words, as information richness increases, populations assess risk more accurately, leading to lower among-population variance in response to manipulations of perceived predation risk. Our results illustrate how explicit consideration of variance in responses can yield important biological insights.

## Results

Using a systematic review, we identified 116 studies that were appropriate for inclusion in our meta-analysis (Supplementary Fig. S1). From these studies, we extracted 645 estimates representing 87 species (Fig. 2A) and 29 countries/regions (Fig. 2B). Using these estimates, we constructed (phylogenetic) multi-level meta-analytic models[23] to

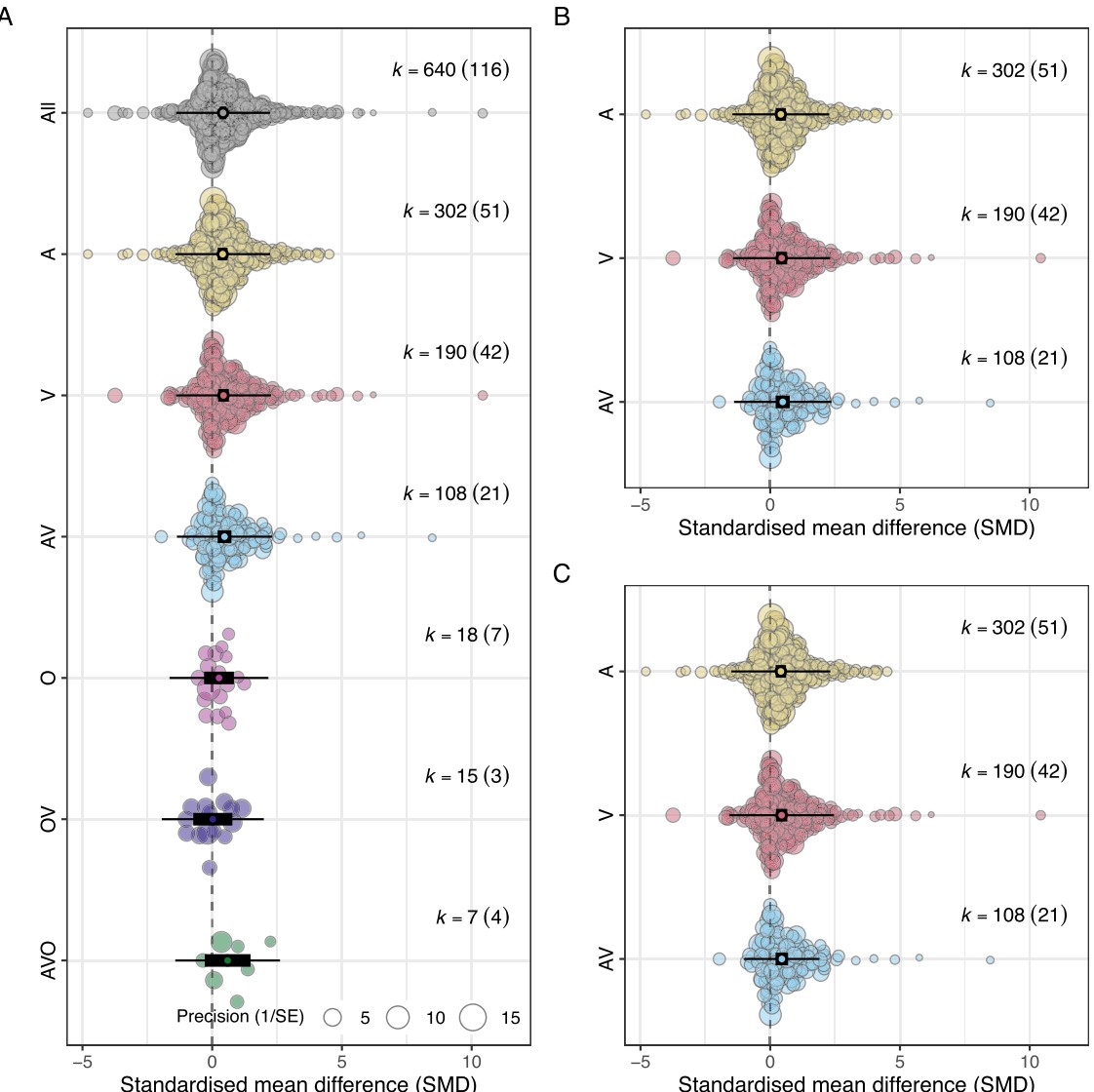

**Fig. 3 | Orchard plot of meta-analytic mean effect sizes, standardised mean difference (SMD or Hedge's g) for each of six treatment levels for experimental manipulations of perceived predation risk: A = acoustic, AV = acoustic + visual, AVO = acoustic + visual + olfactory, O = olfactory, OV = olfactory + visual and V = visual.** The circles denote the meta-analytic means. Note that the black rectangles represent the 95% confidence intervals, while the whiskers denote the 95% prediction intervals. **A** Shows results from meta-analysis including all treatment levels. **B**, **C** Illustrate results from analyses restricted to the three most common cue types (A, V and AV). **B** Shows estimated effects from homoscedastic model, and (**C**) shows estimated effects from heteroscedastic model. Total number of estimates (k) is given on to the right of each plot with the number of studies contributing estimates in parentheses.

understand the effect of different cue types (Acoustic, Olfactory, Visual) and cue combinations on means and variances in responses to manipulations of perceived predation risk in birds. We collected additional meta-data from included studies to assess the role of several putative moderators of the response to manipulations of perceived risk. We also performed analyses to test for publication bias.

Estimates were not evenly distributed amongst the types of unimodal cues or their multimodal combinations (Fig. 3). Most estimates were for experimental manipulations using acoustic cues ($k = 302$), followed by visual cues ($k = 190$), then combined acoustic and visual cues ($k = 108$). A smaller number of estimates were obtained from experimental manipulations of olfactory cues alone ($k = 18$), or olfactory cues in combination with visual cues ($k = 15$) or both visual and acoustic cues ($k = 7$). Within the three treatment levels for which we had a large number of estimated effect sizes (A, V and AV), estimates were relatively balanced across all putative moderators (see Supplementary Information S5; https://itchyshin.github.io/

multimodality/) such that observed treatment effects were unlikely to be due to confounding effects of these moderators.

**Responses to different cues of predation risk: how is information integrated?**

Overall, there was strong support that birds responded in the predicted direction (see Supplementary Information Table S1 for details on coding of predicted effects direction, Fig. 3) to manipulations of perceived predation risk (standardised mean difference, SMD or g = 0.418, 95% confidence interval, CI = [0.288, 0.548]). Total heterogeneity was high ($I^2_{[total]} = 92.82$), phylogeny ($I^2_{[phylogeny]} < 0.01$) species ($I^2_{[species]} = 1.22$) and subject ID ($I^2_{[group]} = 0.00$) accounted for very little variation. Substantial heterogeneity was observed across studies ($I^2_{[across-study]} = 15.58$), with most heterogeneity remaining unexplained ($I^2_{[residuals]} = 75.54$).

As per our a priori assumptions about the level of certainty each cue modality would convey about current predation risk, we first

assessed whether different cue modalities elicited different magnitudes of response. Contrary to our predictions, there was no support that the mean magnitude of response differed as a function of the modality of cue(s) presented (Fig. 3). No pairwise contrasts between treatment categories (types of uni-modal cues or contrast between unimodal and multimodal cues) were significantly different from one another (all $p \geq 0.30$, see Supplementary Information S5; https://itchyshin.github.io/multimodality/ for exact $p$ values for each pairwise contrast. Supplementary Information S5 also includes multi-moderator analyses and sensitivity analyses).

Given the lack of estimates in response to olfactory cues either alone ($k = 18$) or in combination with visual ($k = 15$) and in combination with both visual and acoustic ($k = 7$) cues, we restricted subsequent analyses to estimates derived from the three most common treatment types: A, V and AV. The exclusion of treatments, including olfactory cues (alone or in combination) due to low sample size did not alter the interpretations related to the three most commonly used treatments (A, V and AV). Analyses restricted to the three most common treatment levels reveal that while the mean response to each of these three treatments did not differ, there was strong support for heterogeneous variances (LRT: 18.86, $p < 0.0001$). Specifically, among-study variance in response to acoustic cues alone ($\sigma^2_{[residuals]} = 0.75$) and visual cues ($\sigma^2_{[residuals]} = 0.86$) were comparable in magnitude. However, when acoustic and visual cues were provided together, among-study variance in responses was less than half in magnitude ($\sigma^2_{[within-study]} = 0.35$). This can be seen by the narrower spread (points) of data as well as the narrower 95% prediction intervals (whiskers) for multimodal cues compared with either acoustic or visual unimodal cues (Fig. 3C).

### Exploring the effects of moderators on the responses to manipulations of perceived predation risk

As a secondary analysis, we explored the effects of several potential moderators on the response to experimental manipulations of perceived predation risk. We found that response to manipulations of perceived predation risk varied as a function of the type of response measured (see Supplementary Information Table S1 for definitions and examples of each response type). Specifically, behavioural responses were significantly stronger than physiological responses (estimated difference: $\beta = 0.458$, 95% CI = [0.193, 0.723]), with life-history responses being intermediate in magnitude and not significantly different from either behavioural (estimated difference: $\beta = 0.178$, 95% CI = [−0.062, 0.418]) or physiological responses (estimated difference: $\beta = 0.280$, 95% CI = [−0.025, 0.586]) (Fig. 4A). Responses also varied as a function of treatment duration, with longer treatments eliciting significantly smaller responses ($\beta = −0.046$, 95% CI = [−0.076, −0.015], $R^2_{[marginal]} = 3.43$) (Fig. 4B). However, response type and treatment duration were confounded, making it difficult to disentangle their effects from one another (Fig. 4B).

We also evaluated support for several additional putative moderators. There was no support that additions to the visual treatment (e.g., movement of model predator), setting (lab, field, or semi-natural), season (breeding or non-breeding), study design (within-subject versus among-subject), response period (during or after treatment), control type (blank, control for disturbance, non-predator control), sex of focal individuals (male, female or both), age (adults or nestlings), or predator type (predator to adults, predator to nestlings, or both) on the magnitude of response to manipulations of perceived predation risk (see Supplementary Information S5; https://itchyshin.github.io/multimodality/).

### Publication bias

Visual assessment of funnel plots did not provide evidence for publication bias (Fig. 5A). Results of the Egger regression were consistent with this. The slope of the regression was not significantly different from zero ($\beta = −0.03$, 95% CI = [−0.08, 0.03], $R^2_{[marginal]} = 0.30\%$)

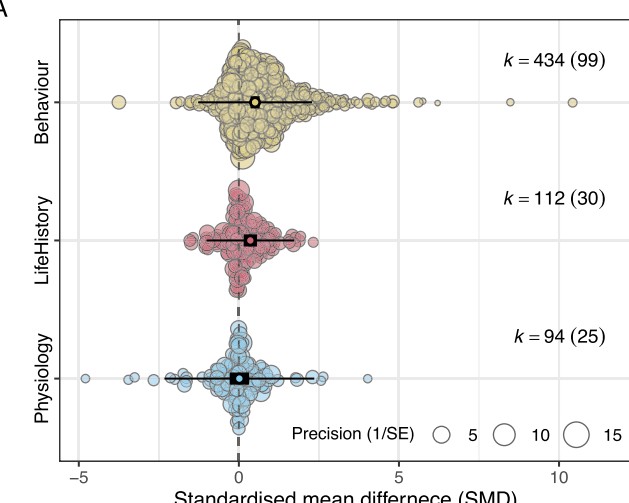

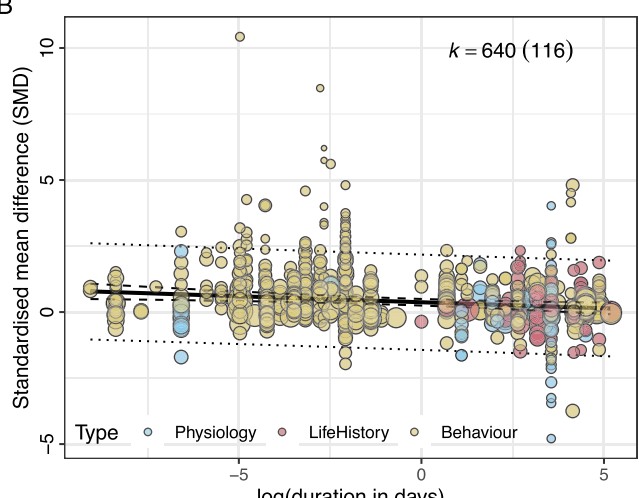

**Fig. 4 | Illustration of the effect of significant moderators of the effect of manipulations of perceived predation risk in birds.** Magnitude of response varies as a function of (**A**) response type (behaviour, life history, or physiology), and (**B**) declines with increasing treatment duration. However, different treatment durations tend to be associated with different response types as shown in (**B**), making it difficult to tease apart their effects. In (**A**), the circles denote the meta-analytic means. Note that the black rectangles represent the 95% confidence intervals, and whiskers denote the 95% prediction intervals. In (**B**), the regression is plotted with 95% confidence intervals (inner dotted line) and 95% prediction intervals (outer dotted line). Total number of estimates (k) is given on to the right of each plot, with the number of studies contributing estimates in parentheses.

(Fig. 5B). We also found no evidence of a time lag effect (Year: $\beta = −0.01$, 95% CI = [−0.03, 0.01], $R^2_{[marginal]} = 0.59\%$ (Fig. 5C; for more relevant results, see Supplementary Information S5; https://itchyshin.github.io/multimodality/).

## Discussion

We used meta-analyses to quantify the effect of experimental manipulations of perceived predation risk in birds on behavioural, physiological and life-history traits and explored the effects of several putative moderators for the relationship. We found strong overall support that birds respond in the predicted direction to manipulated predation risk. However, contrary to our predictions (Fig. 1), we found no evidence that the modality of information about predation risk (acoustic, visual, or olfactory) influenced the mean magnitude of response, nor did combining cues alter the mean magnitude of response (Fig. 3). Interestingly, we found strong support that providing

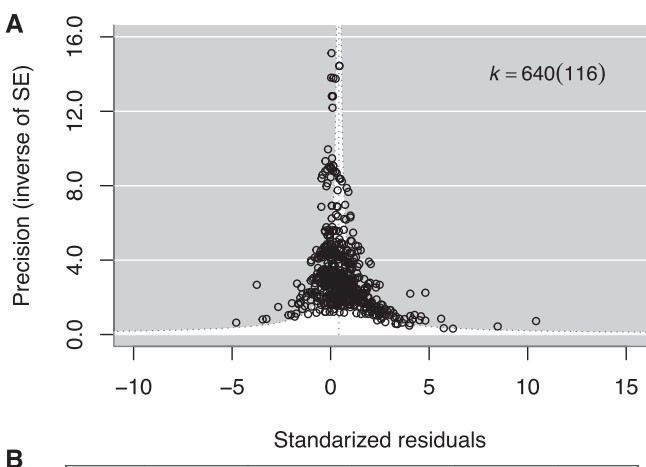

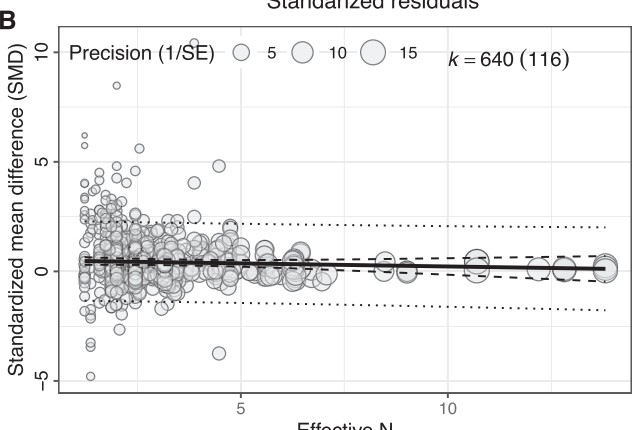

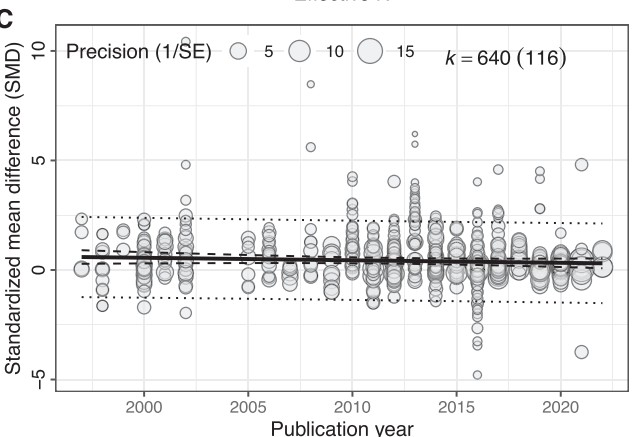

**Fig. 5 | Assessing publication bias. A** Funnel plot. **B** Egger regression to assess funnel asymmetry. 95% confidence intervals are depicted by the two outer dotted lines. **C** Regression to test time lag effect of published effect sizes, with 95% confidence intervals depicted by the two inner dotted lines and 95% prediction intervals depicted by the two outer dotted lines (these are non-linear as the predictions are derived from multi-moderator models). Total number of estimates (k) is given at the top of each plot, with the number of studies contributing estimates in parentheses.

multi-modal cues of predation risk reduced among-study variance in response to manipulations. We discuss the implications of these findings for our understanding of how multimodal cues affect uncertainty and shape animal decision-making in a wide range of contexts.

## Responses to unimodal cues
We assumed that different types of manipulations of perceived predation risk would convey different degrees of certainty about the

current level of risk. Specifically, we assumed that visual cues, such as predator mounts, would provide the highest certainty about the current presence of a predator. In contrast, olfactory cues would provide the lowest level of certainty as these cues can persist even after the predator has left the area. Acoustic cues, such as mobbing calls by conspecifics, were expected to provide an intermediate level of information. On the one hand, they provide social information about current predation risk, but they can be unreliable as they can be given as false alarms[17,18], or may reduce perceived risk as they indicate that the threat is already being attended to[15].

Accordingly, we predicted that the response to visual cues of predation would be greater than the response to olfactory cues, with acoustic cues producing intermediate-level responses. Although the response to olfactory cues tended to be lower compared with either visual or acoustic cues, the 95% CI around the estimated effects overlapped broadly, indicating a lack of support for a difference in response level. The estimated response to acoustic versus visual cues was quantitatively very similar, indicating strong support for no difference. Therefore, contrary to our expectation, risk assessment based on either acoustic or visual cues alone was similar. We suggest this similarity may be because the acoustic cues used were typically mobbing and/or alarm calls of groups of conspecifics (studies using predator vocalisations were rare). Although single individuals may produce false alarms, the risk of a group of conspecifics producing false alarms may be lower. A consensus among group members about current risk (expressed by group mobbing calls) may provide relatively high certainty about current risk such that the response to this social information is, on average, similar to direct, personal information[24]. Unfortunately, we were not able to extract information about the number of individuals present in mobbing call playbacks used in the studies included in our meta-analysis and our suggestion that information certainty for mobbing calls increases with increasing group size requires direct testing.

## Integration of multimodal cues
We were also interested in understanding how access to multimodal cues would shape responses to manipulations of perceived predation risk. There needed to be more studies that used olfactory cues in combination with other cues (olfactory + visual: $K = 3$ studies, olfactory + acoustic + visual: $K = 4$ studies) to allow meaningful analyses of these multimodal cue combinations. However, when comparing responses to either acoustic or visual cues alone versus acoustic and visual cues combined, there was no support for an effect on the mean magnitude of response. This finding is consistent with the notion that the two cues provide redundant information (Fig. 1), which could be expected given that each cue in isolation elicited quantitatively similar responses (Fig. 3). However, our analyses also show that among-study variance in response to multimodal cues was significantly lower compared with responses to unimodal cues (Fig. 3).

This result may be explained by maximum-likelihood estimation (MLE) integration. MLE integration refers to a process by which independent probability distributions are integrated to produce a probability distribution that combines the information from independent estimates[25]. Specifically, if each of independent probability distributions is Gaussian, the combined estimate mean will correspond to the weighted average of the independent estimate means, with the weights being inversely proportional to the amount of uncertainty, or variance, associated with each independent estimate (Eq. 1). Furthermore, the variance of the combined estimate is always reduced relative to either of the independent estimates from which it is derived (Eq. 2). Thus, under MLE integration, responses to multimodal cues are always expected to have lower variance than responses to any unimodal cue

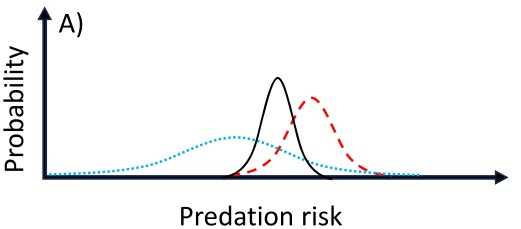

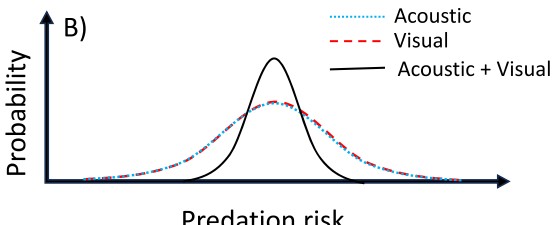

**Fig. 6 | Illustration of multimodal cue integration under two scenarios.** For example, (**A**) an acoustic cue provides a lower mean estimate of risk and higher uncertainty/variance (blue dotted line) compared with a visual cue (red dashed line). The estimated risk that integrates both these sources of information using maximum likelihood estimation (MLE) integration will have lower variance than either alone, and the mean will be closer to the mean of the higher certainty unimodal cue (solid black line). **B** An acoustic cue (blue dotted line) and a visual cue (red dashed line) provide similar means and variances in estimated risk. Under multimodal cue integration using MLE integration (solid black line), the mean estimated risk remains unchanged but has lower variance relative to both unimodal cues.

presented alone.

$$\mu_{AV} = \left( \frac{\frac{1}{\sigma_A^2}}{\left( \frac{1}{\sigma_A^2} + \frac{1}{\sigma_V^2} \right)} \right) \mu_A + \left( \frac{\frac{1}{\sigma_V^2}}{\left( \frac{1}{\sigma_A^2} + \frac{1}{\sigma_V^2} \right)} \right) \mu_V \quad (1)$$

$$\sigma_{AV}^2 = \frac{\sigma_A^2 \sigma_V^2}{\sigma_A^2 + \sigma_V^2} \quad (2)$$

Under MLE integration, the probability densities of predation risk associated with cues presented in isolation yield combined estimates that integrate information about the mean and variance estimations derived from either cue alone[26]. Estimates with less variance are given higher weight under MLE integration so that if the two estimates have different means from their probability distribution, the mean derived through the integration of both estimates will be closer to the mean from the higher certainty cue (Fig. 6A). Importantly, the variance of the combined estimate is always reduced relative to either of the independent estimates from which it is derived. Thus, even if acoustic and visual cues of predation risk have equal means and variances in the probability distributions for estimated predation risk, multimodal cues that combine information from visual and acoustic cues will still have lower variance than either unimodal cue alone (Fig. 6B). However, cue integration occurs at the level of individuals. Thus, under MLE integration, we would expect a reduction in among-individual variance when combining cues with equal probability distributions[26]. Assuming different study populations had access to the same cues with the same probability distributions, we would not predict MLE integration of multimodal cues to lead to a reduction in among-study variance (Fig. 6).

However, we argue that the assumption that the probability distributions of cues used across studies are identical is unrealistic for several reasons. First, even within cue types, studies vary in numerous features that are likely to affect risk assessment. For example, we found that response magnitude was affected by treatment duration, with longer exposure to cues resulting in smaller responses (Fig. 4B). Furthermore, numerous studies have shown that the same species of predator can elicit different responses depending on postural cues about the current threat level and/or the distance at which the predator is first detected (e.g.,[6,9,27–30]). Such variation also exists among studies and may be expected to contribute to among-study variance in response. For field studies, particularities of the study site, including habitat features that affect the ability of birds to detect or evade predators, year-specific environmental conditions that affect the risk of energy shortfall, or among-study differences in population size that influence dilution of predation risk, among others, are all likely to have biologically important impacts on perception of predation risk[1,31]. Thus, we can expect large among-study variance in risk assessment

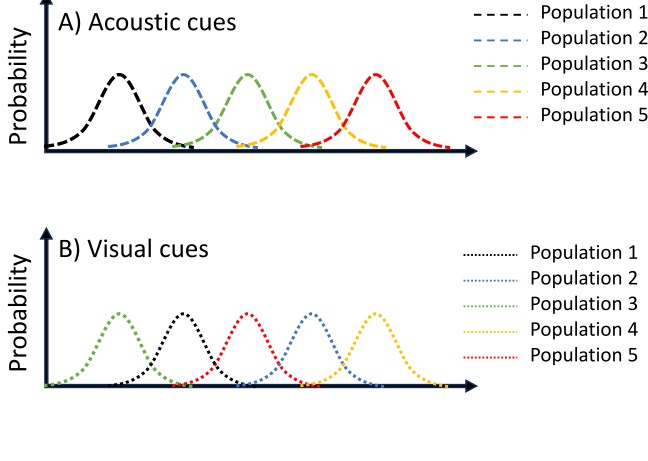

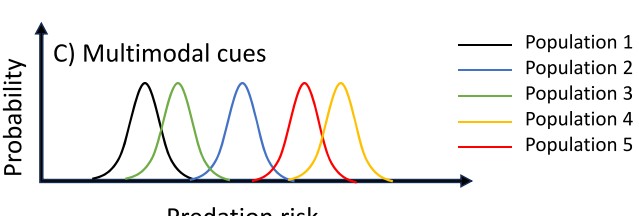

**Fig. 7 | Illustration of how maximum likelihood estimation integration (MLE) could result in lower among-study variance in response to manipulations of perceived predation risk when two redundant cues are integrated relative to the among-study variance when either cue type is presented alone.** Each panel illustrates the same five hypothetical populations (shown in five distinct colours). If there is across-study heterogeneity in the probability function associated with study-specific unimodal cues, as shown in (**A**) (Acoustic) and (**B**) (Visual), then even if the mean and among-study variance in response to each of the two unimodal cues are identical, maximum likelihood integration will result in lower among-study variance, as shown in (**C**).

even when the same cue modality is used (Fig. 7A, B). Indeed, our analyses support this interpretation because study ID accounted for substantial heterogeneity among estimates. Under MLE integration, high among-study variance in the perceived risk associated with a given cue type would result in similar mean responses to unimodal versus multimodal cues but lower among-study variance under MLE integration (Fig. 7C).

## Moderators of the effects of perceived predation risk in birds
We also explored the effects of several putative moderators on the magnitude of response to manipulations of perceived predation risk. Surprisingly, several moderators previously described as important were not found to affect the magnitude of response in the present meta-analysis (Supplementary Information S5; https://itchyshin.github.io/multimodality/). Specifically, we found no evidence that

adding movement to the visual manipulations of predation risk, such as side-to-side head movement or gliding movement by the model predator, consistently affected the mean response significantly. Similarly, whether the control treatment was a blank control, a control for the disturbance associated with the predator treatment, or an equivalent stimulus from a non-predator species did not systematically affect mean response levels. Both of these findings are at odds with results from earlier empirical studies showing that birds can exhibit graded responses to stimuli representing differing levels of risk and/or disturbance[6–8]. Given the sample sizes for each of the moderators considered, (see Supplementary Information S5; https://itchyshin.github.io/multimodality/), we would have had the statistical power to detect general effects of these moderators if they were present. The fact that we did not detect significant effects for several of the moderators considered highlights the context-specificity of ecological field studies, which makes both exact and partial replication challenging[32]. We suggest that the lack of effects reported here again can be attributed to high among-study variance such that uncontrolled among-study variance had a larger impact on response to manipulations of perceived predation risk than specific features of the experimental treatment (e.g., predator posture or type of control).

In fact, only two of the explored moderators had detectable effects on mean response to manipulations of perceived predation risk: duration of treatment and response type. Longer treatment durations were associated with significantly smaller effect sizes (Fig. 4B). This result is consistent with the notion that animals may habituate to cues that are presented repeatedly for extended periods of time[33], or that lower allocation to predator avoidance is adaptive when high-risk situations are frequent and/or lengthy[34]. However, another possibility is that researchers design their studies based on expected responses, such that traits that are expected to exhibit small and/or slow responses to manipulations of risk are typically studied using experiments with longer treatment durations. Indeed, we found that treatment duration was the longest for studies investigating life history responses to perceived predation risk, followed by studies of physiological responses, with studies of behavioural responses tending to have the shortest duration (Fig. 4B). While our analyses did detect an effect of response type on response magnitude, with behavioural traits exhibiting the largest effect sizes (Fig. 4A), because response type was confounded with treatment duration, we cannot conclusively tease apart their effects from one another.

### Limitations and future directions
Our meta-analysis revealed significant heterogeneity in responses to manipulations of perceived predation risk, with most heterogeneity existing at the level of the observation (i.e., single estimates), followed by study ID. This indicates that responses are context-specific and that among-study variance in ecological context and particularities of how treatments were carried out have important consequences for how birds respond to experimental manipulations of predation risk. For example, the information conveyed by playbacks of mobbing calls may depend on the number of callers that can be heard in the playback. Importantly, there were several limitations to the available data. First, the lack of studies that included olfactory cues, either alone or in combination with other cues, meant we could not evaluate whether integration of information of olfactory cues differed from other cue modalities. The lack of studies considering olfactory cues of predation risk in birds may be a result of the common misconception that birds have a weak sense of smell[21]. Although there is now ample evidence that most birds do in fact have a strong sense of smell[35], more empirical studies are needed to test whether birds exhibit systematically lower responses to olfactory cues. In contrast, the use of olfactory cues to assess predation risk in aquatic systems has been studied extensively[19,20]. The relative information value of different cue types may differ across different environments and different taxa. For

example, the information value of chemical cues in aquatic systems may be fundamentally different compared to terrestrial systems due to differences in how cues persist in these different environments. Thus, meta-analysis of studies evaluating unimodal and multimodal cue integration in other systems and taxonomic groups would provide further insights on whether and how cue integration leads to uncertainty reduction in risk assessment. Additionally, we did not detect a phylogenetic effect of species on responses to perceived predation risk. However, Passeriformes in particular, were over-represented within the studies included in the meta-analysis (Fig. 2A), contributing 84% of all estimates (542 out of 645) despite representing about 66% of all bird species. This may have limited our power to detect phylogenetic effects. Expansion to include other non-avian taxa would be critical to assess the generality of MLE integration in animal decision making in anti-predator contexts.

Further, while our meta-analysis did synthesise studies from 29 countries from five continents, the representation was heavily skewed towards North America and Europe (Fig. 2B). Given that our analysis indicates an important effect of study ID, which we presume is due to study-specific context (e.g., baseline predation risk, flock size, food availability, ambient conditions, etc.), a more balanced global representation of studies would help ascertain the generality of our results. Finally, at least two potential moderators of the effect of manipulations of perceived predation risk on birds were confounded in our available dataset; treatment duration and response type. More studies employing relatively short-term manipulations of perceived predation risk to investigate physiological and life-history responses are needed to better understand the causal effect that each of these moderators (treatment duration and response type) exert independently.

We found no evidence that the type of unimodal cue affected mean response, nor did multimodal cues differ in mean response compared to unimodal cues (Fig. 3). However, there was strong support that among-study heterogeneity was lower for responses to multimodal cues compared to unimodal cues. This finding is consistent with maximum likelihood estimation (MLE) integration. Importantly, the MLE integration hypothesis applies across multiple levels of biological organisation, including cue integration at the level of individuals, populations and studies. A logical next step to formally test this hypothesis would involve manipulating unimodal and multimodal cues in different contexts (e.g., predation risk, mate choice, etc.) and across different scales (within-individuals, among-individuals within the same population and across studies) to test (1) whether multimodal cues lead to lower variance in responses across each of these scales as predicted by MLE integration and (2) the generality of MLE for information integration problems.

Our meta-analysis shows that providing two complementary cues indicating predation risk does not alter mean responses but leads to lower among-study variance in response. Our finding demonstrates that explicit consideration of variance can yield important biological insights[36,37]. Based on these meta-analytic insights, we outline a framework for cue integration that incorporates the effects of cue integration on both means and variances in response: maximum likelihood estimation (MLE) integration. Although the MLE framework has been shown to apply to the integration of visual and haptic cues in humans[26], to date, studies of cue integration in non-human animal systems have not explicitly considered the impact of cue integration on variance in responses. Given that MLE integration can apply at different scales, from individuals to populations, it may be relevant to understanding information integration in animal decision making in a wide range of contexts.

## Methods
### Literature search and inclusion/exclusion criteria
We followed the steps outlined in the Preferred Reporting Items for Systematic Review and Meta-Analysis (PRISMA) protocol[38] for our

meta-analysis as recommended by Nakagawa and Poulin[39]. We additionally verified the reporting of our study items using the PRISMA-EcoEvo guidelines outlined in ref. [40]; see Supplementary Information Table S2. We performed our literature search in the online databases Web of Science (All databases) and Scopus accessed through the University of Alberta libraries subscription. We had search terms related to predation, experiments and taxa. The predation-related search terms used were: 'predat* risk' OR 'pred* danger' OR 'perceived predat*' OR 'perceived risk'. The experiment-related search terms were 'experiment*' OR 'manipulat*' OR playback* OR treatment*. Because our meta-analysis was restricted to birds, we used the additional taxa-related search terms: 'bird*' OR 'aves'. We searched for articles using these terms in the 'Topic' field. Articles had to include at least one of the search terms from each of the three topic strings.

JDAT, NK and KJM conducted the initial scoping review, developed search terms and defined inclusion/exclusion criteria. The final literature search was conducted on February 18th, 2022. Our search criteria produced a total of 814 unique references (Supplementary Information Fig. S1). As a first step, we screened these references by title and abstract to assess their relevance to the meta-analysis. Title and abstracts were screened by four observers (RAM, SS, DMH and KJM) independently using Rayyan[41]. Any disagreements were resolved through joint discussion. This resulted in a total of 171 articles for which the full text was read by JDAT or KJM. To be included in the meta-analysis, studies had to fulfil each of the following criteria:

1. The study had to present an experimental manipulation of perceived predation risk. Manipulations of perceived risk included experimentally providing cues of predator presence (olfactory, visual or auditory cues), social cues of predation risk (e.g., mobbing calls or alarm calls), or any combination of the above. For olfactory cues, we considered both presentation of chemicals obtained directly from predators (e.g., by exposing birds to material that had been housed with predators and absorbed predator odours) as well as synthetic predator odours (e.g., commercially available chemical compounds that are naturally present in predator anal gland secretions, faeces, or urine). For acoustic cues, we only considered vocalisations made by known predators or vocalisations made by the focal species (e.g., mobbing or alarm calls). We did not include studies that aimed to test whether a cue was recognised by birds (e.g., presentation of novel predator or evaluation of social learning about predation risk). We did not consider the presence of human observers alone as an experimental manipulation of perceived predation. Similarly, we did not consider mobbing or alarm calls produced in response to humans as a relevant manipulation of perceived predation risk. We included studies that manipulated perceived predation risk using live predators as long as the presence/absence of the predator was determined experimentally (e.g., caged predator, or presented via falconer).
   Studies that manipulated predation risk without providing cues related to the presence of actual predators were not included. For example, we excluded studies that manipulated the size of the nest box entrance so that some were accessible by predators and others were not, or studies that manipulated landscape features (e.g., distance to obstructive cover, distance to protective cover) that alter the ability to detect and/or evade predators. We also did not include studies that manipulated predation risk using predator removals or exclusions, as these did not report the predator cues (type, frequency) that were encountered in the control groups (i.e., non-removal plots or outside exclusions).

2. The study had to provide data on behaviour, life history, or physiology/morphology as a function of manipulated perceived risk. The full list of traits included in the meta-analysis and their definitions is provided in Supplementary Information Table S1.

3. The study had to allow for the calculation of effect size for a behavioural, life history, or physiological variable in response to a manipulation of perceived predation risk as described in (1). The study had to include a control for the manipulation, such as data on the response variable prior to the experiment in the same set of individuals (Before-After-Control-Impact (BACI) or within-subject design), or contrasts between sets of individuals exposed to the manipulation and individual not exposed to the manipulation (among-subject design). Studies that only contrasted different manipulations of perceived risk (e.g., response to visual cue versus response to acoustic cue) were not included. We excluded any estimates for which there were less than $N = 3$ individuals in a given treatment group because the standard deviation (SD) could not be estimated well with small sample sizes (see below calculation details).

4. The study had to be conducted on birds and present species-specific results. Studies that presented mixed-species responses (e.g., the average response of a mixed-species flock) were not included in the meta-analysis.

5. We initially considered any behavioural, life history, or physiological trait if the study fulfilled the four criteria listed above. However, following full-text screening of all articles, we removed studies/estimates if there were not at least $N = 3$ studies that provided extractable data for that response variable.

These selection criteria resulted in a total of 113 papers that were appropriate for inclusion in our meta-analysis[15,42–153], and five data sets associated with these studies that were archived on Dryad, a public data repository[154–158]. Studies that were deemed not to fulfil these selection criteria ($N = 58$) upon reading the full text are listed in Supplementary Information Table S3, along with the reason for their exclusion. We additionally included $N = 2$ articles not captured by the search criteria but known to the authors to be relevant[159,160], and $N = 1$ article that was rejected based on title/abstract but which was known by the authors to include relevant data[161], resulting in a total of 116 articles from which we extracted estimates. The full PRISMA flow chart is provided in Supplementary Information Fig. S1.

## Data coding and calculation of effect sizes

For each estimate extracted, we noted a number of variables for assessing publication bias (e.g., time lag effect) and exploring putative moderators. These included: (1) the year the study was published to allow us to investigate the time lag effects of published effect sizes (see *Publication bias*, below), (2) the species name of the focal organism to allow us to control for phylogeny in the meta-regression, (3) whether the experimental manipulation of perceived predation risk involved cues of a single predator species (and if so, the predator species name) or multiple predator species, (4) the guild of predator(s): bird, mammal, fish, reptile, not specified, or multiple guilds, (5) whether the predator was a predator of adult birds (A), nests (including eggs and nestlings, N), or both (B), (6) the setting of the study: field, lab, semi-natural (e.g., wild-caught birds held in outdoor aviary), (7) the treatment: A = Acoustic, O = Olfactory, V = Visual, or any combination of the above, (8) the season (breeding, non-breeding), (9) the type of comparison: among = among individuals, cohort comparisons; within = within-subject comparison such as before/after, (10) treatment duration, expressed as number of days. Treatments conducted within a single day were coded as the proportion of the day that the treatment lasted, assuming a 12 h daylength, (11) control type: blank = no experimental control (e.g., before-after study design), NonPred = non-predator control, disturbance = control for the disturbance associated with the treatment or non-biological components of treatment such as the presence of a speaker, (12) sex of focal individuals: male, female, both (includes studies that explicitly stated both sexes were included, as well as studies which made no explicit mention of sex of focal

subjects), (13) age of focal individuals when treatment was applied: A = adults, N = nestlings, J = Juveniles, E = eggs. Detailed rationale for collecting each of these variables is provided in Supplementary Information Table S4.

We collected relevant sample statistics (e.g., mean, median, sample size, standard deviation, standard error, quantile range, etc.) for responses to control and treatments from each study or its associated data repository. When the relevant data were presented in figures, we extracted the data using WebPlotDigitizer 4.1[162]. and transformed relevant study results into a standardised effect size (SMD, or often referred to as Hedge's g). Effect sizes and variances cannot be calculated when proportion responses include either 0 or 1 (e.g., proportion of nests abandoned). Thus, we replace 0 proportion responses with 0.025, and 1 proportion responses with 0.975 following Fox and Weisberg[163]. We used Hedge's g as our standardised effect size because we were interested in the effect of categorical variables (predation risk treatment) on behaviour, life history and physiology and this effect size removes bias for small sample sizes that occur when using other effects sizes such as Cohen's d.

In order to allow us to estimate global effects in our meta-analyses, all variables were coded so that effects in the predicted direction were positive. This means that estimates for categories where the predicted effect was 'Decrease' were all multiplied by -1 prior to analysis. For example, inter-scan interval and avoidance of protective cover are both predicted to decrease with increasing predation risk. Thus, estimates for these response variables were multiplied by -1 prior to analysis. A complete list of response variables, direction of predicted effect, and re-coding of effect direction is provided in Supplementary Information Table S1.

## Meta-analysis and meta regression analysis

We conducted all statistical analyses including exploratory data analyses in the program R version 4.2.3[164]. We calculated standardised effect sizes and their sampling variance using a custom function that converted SMD (Hedge's g) calculated via the effect size calculator at the Campbell Collaboration website (see Supplementary Information S5: https://itchyshin.github.io/multimodality/#custom-functions). Using these, we constructed (phylogenetic) multi-level meta-analytic models[23]; we used the rma.mv function in the R-package metafor[165] along with the R-package MuMIn for multi-model inference[166]. The meta-analytic models were to ascertain that, overall, birds responded to treatments compared to control conditions.

Initially, our meta-analytic model had five random effects that were considered a priori to be potentially important sources of variation and non-independence in estimated effect sizes. These were: (i) the phylogenetic effect of species, (ii) species identity (a non-phylogenetic component of species), (iii) group (i.e., a unique set of individuals to account for the fact that the same individual could be used to estimate multiple effect sizes), (iv) study ID (i.e., a unique study identifier to account for non-independence between estimates derived from the same study population), and (v) observation id (i.e., a unique id value assigned to each effect-size estimate, equivalent to the residual term in a normal linear model). We obtained the avian phylogenetic tree from ref. 167. To account for phylogenetic uncertainty, we used 50 posterior samples of the avian phylogenetic tree and merged results using Rubin's rules according to Nakagawa and DeVillemereuil[168]. Because phylogeny played little role in this analysis, we report results from one tree in Results below (see also Supplementary Information S5; https://itchyshin.github.io/multimodality/#meta-analysis).

These random effects did not account for all non-independence among sampling variances (i.e., correlations due to the same individuals being used to obtain more than one effect size)[169]. To deal with this, we created a variance-covariance matrix to add to meta-analytic models by assuming sampling variances from the same cohorts

(subject ID) from the same studies have the correlation $r = 0.5$, as suggested by ref. 170. For meta-analytic models, we calculated the multilevel-model version of heterogeneity ($I^2$), which quantifies variance not due to sampling error, for each random effect and the total heterogeneity following Nakagawa and Santos[23]. Based on these analyses, only species identity, record ID and observation ID were retained. For subsequent analyses (i.e., meta-regressions), we dropped the phylogenetic effect of species and group ID as these accounted for <0.01% of the heterogeneity.

To explain the observed heterogeneity ($I^2$), we created a set of meta-regression models. The moderators considered were: cue modality, trait type (behaviour, life history or physiology), treatment duration (in days), sex of the focal individual (male, female, or both), type of predator used (i.e., whether the predator targets adults, eggs/nestlings or both), predator guild, study design (within-subject versus among-subject), season (breeding versus non-breeding), setting (field, lab or semi-natural) and control type (blank, disturbance control, or non-predator control).

To address our main question, we first tested for the effect of cue modality. We did this in multiple steps. We first constructed a model including all six treatment levels for which we had estimates: acoustic (A), visual (V), olfactory (O), acoustic + visual (AV), olfactory + visual (OV), acoustic + visual + olfactory (AVO). For completeness, we present the estimates for the three cue types combined despite not having a strong a priori prediction. However, because there were few estimates for treatments involving olfactory cues either on their own or in combination with other cue types (see Results), we also constructed models that were restricted to estimates from studies based on A, V and AV treatment levels. Although we did not have a priori predictions regarding how multimodal cue integration would affect variance in response to manipulations of perceived predation risk, visualisations using orchard plots revealed a clear difference in variability among different treatment levels[171,172]. Therefore, we considered both homoscedastic and heteroscedastic models, as recommended by ref. 173.

As a secondary analysis, we considered all the other moderators above and where appropriate, we considered both homoscedastic and heteroscedastic models (see Supplementary Information S5; https://itchyshin.github.io/multimodality/). For all models, we assessed the importance of moderators by calculating marginal $R^2$ sensu[174]. We visualised meta-analytic results as well as other relevant results mainly using the R packages ggplot2[175], orchaRd[171,172], ggalluvial[176] and ggtree[177]. Data and reproducible analyses are provided in Supplementary Information S5 (https://itchyshin.github.io/multimodality/).

## Publication bias

We evaluated evidence for publication bias by assessing funnel plot asymmetry and tested the significance of the asymmetry using a multilevel version of Egger's regression[178]. We included the square root of the effective sample size (effective $N$) as a fixed effect in Egger's regression and also included the following random effects based on the variables that contributed most to heterogeneity in the null model described above: species ID, study ID and observation ID. We assessed the presence of a time lag effect, which occurs when larger or statistically significant effects are published earlier compared to small and/or statistically non-significant effects[178]. To do this, we regressed standardised effect sizes (Hedge's g) against publication year[179,180], also known as a decline effect[181], with the same random effects as Egger's regression model (species ID, study ID and observation ID). Furthermore, we conducted a leave-one-study-test to see whether a particular study had a major impact on the overall effect (see Supplementary Information S5; https://itchyshin.github.io/multimodality/).

## Reporting summary

Further information on research design is available in the Nature Portfolio Reporting Summary linked to this article.

## Data availability

The data generated in this study are archived in the Open Science Framework (OSF) project (https://osf.io/9vmzx/)[182]. All data required to reproduce the analyses and figures presented in the manuscript are available to use under the following licence: CC-By Attribution 4.0 International. The data are available for the main analysis is provided in the file 'dat_19_07_2023_spp.csv'. Descriptions of all meta-data are provided in 'Meta-data.csv', and data related to study species and study country required to generate figures is provided is 'Species list.xlsx' and 'StudyCountries.xlsx', respectively.

## Code availability

All code required to reproduce the analyses and figures presented in the manuscript are archived in the Open Science Framework (OSF) project (https://osf.io/9vmzx/)[182]. The code can be found in the file called 'Multimodal_MA_scripts.R'.

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

## Acknowledgements

This work was supported by the Canada Research Chair Program and an NSERC Discovery Grant to K.J.M. (RGPIN-2018-04358). S.N. was supported by the Australian Research Council Discovery Grant (DP210100812), and a part of this work was done during his visitorship to the Okinawa Institute of Science and Technology (OIST) through the Theoretical Sciences Visiting Program (TSVP).

## Author contributions

J.D.A.T. and K.J.M. conceived the study. J.D.A.T., N.K., A.B., D.M.H., R.A.M., S.S. and K.J.M. contributed to the literature review. J.D.A.T. and K.J.M. performed data extractions. K.J.M. wrote the manuscript. S.N. performed the data analysis and wrote related materials. All authors contributed to revisions and approved the final version of the manuscript.

## Competing interests

The authors declare no competing interests.
