## [Peer Review File · Nature Communications]

A systematic review and meta-analysis of unimodal and multimodal predation risk assessment in birdsREVIEWER COMMENTS

Reviewer #1 (Remarks to the Author):

Review of: Mathot et al. Integration of multimodal cues does not alter mean...

Review by: Dan Blumstein

The authors conduct a exemplary systematic review and meta-analysis to study multimodal risk assessment in birds. They find some evidence of redundancy but argue that to properly understand highly variable responses, future researchers should pay more attention to the variance in responses and use a maximum-likelihood framework to interpret their data. This is a very important contribution and clearly illustrates how meta-analyses should be conducted.

L108ff: Nice predictions. I would expect that multiple callers (e.g., like those engaged in active mobbing) might provide more information about risk than a single caller—either giving an alarm call or a single individual's bout of mobbing calls. And yes, seeing an individual being mobbed might communicate lower risk because the predator is being 'taken care of' by others. However, we're empiricists and your results will tell us what birds (at least the studied ones in your sample) actually do.

Methods:

Exemplary! The data/code in the GITHUB site are amazing and oh-so-useful.

Discussion:

Super interesting that there's no real difference between the response to acoustic and visual cues. This alone was worth the analysis!

The MLE section is really interesting and the focus on reduced variance after being exposed to multi-modal stimuli reasonable. I think the authors make a great case for why in such variable systems this is a reasonable way to frame the interpretation of multimodal risk assessment.

I think an important source of variance is the time duration under which the response is measured: behavioral responses are immediate, but other physiological and life history responses are not. If I understand the results (in the supplement) correctly, physiological responses are less than behavioral responses (which I assume are the reference category) but life history and behavioral responses do not differ. I find this weird. Presumably the temporal order of responses is behavior (immediate), physiology (soon after), life history (longer after). Thoughts? What would happen if the same analyses were run separately on the different types of responses (behavioral, physiological, and life history)? There seem to be sufficient sample sizes...

Interestingly, there were no interactions with modality...

However, looking at the results, why are there both male and female estimates given I would assume one would be the reference category (as seen above with behavior being the reference category)? Is there a problem in your data set? Where am I going astray in interpreting these results?

Reviewer #1 (Remarks on code availability):

The code (which I did not run) looks well annotated.

Reviewer #2 (Remarks to the Author):

The authors examined how predation cues presented alone or in combination can alter responses to predation risk in birds at many levels including behavioural and physiological. This topic has

attracted much attention as responses to predation risk are fundamental in many taxa. This is a timely review of such effects and as far as I can tell the analysis was performed adequately.

The results are a little underwhelming in the sense that overall the effect sizes were rarely different from 0 suggesting that researchers must rethink how they manipulate predation risk. This was not mentioned in the paper. The predictions regarding ranking of uncertainty according to cue type are rather simplistic, but this might not be the authors' fault as it is pervasive in the literature. I agree with the authors that the current thinking, which focuses only on mean responses, needs refining. This being said, it was not clear what the paper was really proposing. The finding that variance in effect sizes varied depending on the combination of cues is interesting (although it was hard to judge from the figure) but why the authors looked at this was never explained. The idea was developed in the discussion making it difficult to really appreciate what it might mean. The paper works best when looking at the evidence based on the mean and would need an overhaul if the idea of variances is presented.

It is a pity that the authors did not look at the evidence from other taxon beyond birds given that they are targeting a high-ranking journal with a broad audience. I have other minor comments listed below.

Line 79: Are the results of a review and meta-analysis really a test of predictions? It seems to me that it is more an assessment of the evidence for or against predictions, not a test proper. Each included study is a test as it uses experimental manipulations of cues to examine responses to risk. The meta-analysis does not do that. Perhaps consider rephrasing this.

Line 83: I am a little surprised to see olfactory cues in birds. I can certainly see how useful they can be for mammals but I thought that most birds were not really using such cues except birds like vultures. Can you elaborate on this or was this just an example?

Line 85: What about predator calls rather than calls by prey in response to a predator? Are such calls also considered less certain than visual cues? Please elaborate about predator calls, at I think such calls are used quite frequently in the literature. Contrary to a predator mount, for instance, a predator call suggests that a predator is present and also active. Not sure such calls would have less certainty than visual cues. Alarm calls by prey species are probably a little uncertain given that without visual cues they might be false alarms. Not knowing which calls were used in these experiments makes it difficult to be categorical about the rankings.

Line 99: Again the meta-analysis is not really making predictions but looking at the evidence in support of predictions from the primary literature.

Line 111: So again only mobbing calls are considered? I would think that a visual cue along with a predator call would lead to cue enhancement in your framework. Please explain these choices.

Line 129: Not all papers use birds or aves in the topic section. For instance, in a bird journal, it would be redundant (no pun intended) to add birds in the title, abstract, key words or topic. In your experience, was this not an issue?

Line 184: What do you mean by associated datasets? I thought it was published papers that were reviewed. Please clarify.

Line 187: Known to the authors. I can understand the logic of this but then again what if more were missed given my previous comment? I doubt that many have been missed. Given that you are the experts one can assume that this is probably a small proportion.

Line 191: I appreciate the effort to get as much information about each included study, but I was not always clear what was the motivation behind the collection of data for many variables (e.g. single vs multiple predators, stage of predation). All this information probably matters, but we were not told why it matters. This needs some justification.

Line 240: I was not clear about what observation id was. Can you elaborate?

Line 242: Did you consider using a consensus tree instead of using one arbitrary tree?

Line 248: This procedure seems to assume that all studies used the same subjects repeatedly. This is not necessarily the case. What does it mean for such studies when including this variance-covariance design?

Line 267: What is the prediction regarding the triple combination of cues? This was not presented in the introduction.

Line 288: For the less knowledgeable readers amongst us, can you elaborate on the idea of time lag. I am familiar with funnel plots but less so with other types of analyses. This journal is for a non-specialist audience.

Line 299: I am curious as to what type of olfactory cues were used in birds, but as I thought this was not frequent at all.

Line 307: Judging from figure 4, none of the CIs exclude the value of 0. I was not clear what was the overall effect referred to here.

Line 332: I must admit that the lower variance is not at all obvious from this figure.

Line 340: Can you describe what a life-history response might be?

Line 373: This idea that variance in responses might vary depending on the treatment combination was not explained in the introduction. As presented here, it almost appears as a post-hoc finding.

Line 385: To return to my previous point, it would seem that a predator call might be just as relevant as a visual cue alone.

Line 392: Can we really have strong support for no effect? The evidence to me suggests that the evidence for a difference is weak.

Line 395: To reiterate my point, perhaps it would be helpful to use type of acoustic cue used as a moderator variable or at least to present a breakdown of the different types of acoustic cues used.

Line 412: This topic of MLE integration should have been introduced earlier as it seems very relevant to the ideas tested here. I got the impression that this was added following the results. Given the complexity of the ideas expressed here, more than a cursory treatment is needed.

Line 506: Passerines are very numerous. Can this alone explain why they appear more commonly in such studies?

Reviewer #3 (Remarks to the Author):

In this paper, the authors perform a meta-analysis to test whether and how uni-modal versus (simultaneous) multi-modal cues of potential predation risk affect anti-predator responses among birds. I was actually quite impressed with the number of studies that have performed this type of experiment and so I think this meta-analysis does a really nice job of summarizing this literature and providing some concrete conclusions and considerations for future work. I would now actually be quite excited to see a similar meta-analysis done on another taxonomic group (my guess would be fish would be the next biggest set of studies) as I could imagine that we might expect the cue responses to be similar in some regards (visual cues probably offer highest certainty) and different in others (olfactory cues might offer far more information in aquatic systems, especially given that many studies have manipulated both predator chemical cues and alarm cues from injured prey). But anyway, that's just an idea for future work and I only mention it because it speaks to how interesting I found this paper and that it spurred my own thoughts on future work quite a bit!

But altogether, my impression of this paper is very positive. It's well-written and easy to follow. The authors appear to have followed all the most up-to-date guidelines in how to perform systematic and reproducible meta-analyses (e.g. explicit flowcharts showing inclusion/exclusions criteria; PRISMA checklist; accessible code etc). As someone who's also done several meta-analyses themselves, this all looks completely in order and a great example of clear reporting of their methods.

So only real comments are in some areas where I think a bit more detail or expansion might be useful.

1 – I wonder if it might be useful to add another paragraph or so to the introduction that really expands on how cues might differ in the expected change in the magnitude of the response based on the intensity of the cue, and the certainty of the cue. Both of these topics are brought up in the first paragraph of the intro together. I'm pretty well steeped in the cue certainty literature myself so I quickly caught the difference between these aspects of cue information, but a more naïve reader might benefit from a bit more detail about how these axes differ from each other and how they combine.

2 – In the results on Line 306, the authors state that "birds responded in the predicted direction" which then made me realize that I don't think I saw any mention in the methods about exactly how these directions were coded from the data extraction? First off, what is the predicted direction (increase or decrease)? And who predicted this – did the authors make these predictions and so code the responses themselves, or did they follow whatever the authors of the included studies predicted? I think either way is likely fine, but it should be mentioned. Would be useful to have some more detail here about how to interpret these: which types of behaviors/physiological responses were expected increase versus decrease in response to predation threat? I don't think this affects the interpretation at all, but some more biological detail would be nice.

3 – I find the mention of maximum likelihood estimation as an interpretation of the variance results very interesting. Reading through this immediately made me think though about why would we expect to see lower variance at the among-SPECIES level if MLE is clearly being done at the individual level. But I was then happy to see that authors anticipated this and had a clear discussion of how these two levels may interact and so completely answered my question. My guess is that this result (that audio plus visual cues results in lower study variance) will actually be very informative for future work as it clearly demonstrates that presenting both cues together is more likely to elicit the appropriate response from the species than either presented alone. My guess is this paper should be heavily cited as a justification for this method in the future.

Reviewer #3 (Remarks on code availability):

I checked the github page they cite in their paper, which nicely presents all of their results (including code and figures). I didn't run the code on my own computer, but quickly read through it online and it looks well documented and the figures match the paper which is always a good sign.

RESPONSE TO REVIEWERS' COMMENTS

Reviewer #1 (Remarks to the Author):

Review of: Mathot et al. Integration of multimodal cues does not alter mean...

Review by: Dan Blumstein

The authors conduct a exemplary systematic review and meta-analysis to study multimodal risk assessment in birds. They find some evidence of redundancy but argue that to properly understand highly variable responses, future researchers should pay more attention to the variance in responses and use a maximum-likelihood framework to interpret their data. This is a very important contribution and clearly illustrates how meta-analyses should be conducted.

Response: Thank you!

L108ff: Nice predictions. I would expect that multiple callers (e.g., like those engaged in active mobbing) might provide more information about risk than a single caller—either giving and alarm call or a single individual's bout of mobbing calls. And yes, seeing an individual being mobbed might communicate lower risk because the predator is being 'taken care of' by others. However, we're empiricists and your results will tell us what birds (at least the studied ones in your sample) actually do.

Response: This is a good point- the number of individuals participating in a mobbing call likely affects the way receivers value the information. Unfortunately, studies using mobbing call playbacks don't report the number of individuals that can be heard in their recordings, so we cannot address this analytically. However, we have added detail to the text highlighting that this may be an important source of heterogeneity in the treatments that involve mobbing playbacks (lines 525-527).

Methods:

Exemplary! The data/code in the GITHUB site are amazing and oh-so-useful.

Response: Thank you!

Discussion:

Super interesting that there's no real difference between the response to acoustic and visual cues. This alone was worth the analysis!

Response: Thank you. We were also surprised by this result, but agree that it is an important insight, particularly given how well powered the meta-analysis was (i.e., the lack of effect is not due to low power, not a false negative).

The MLE section is really interesting and the focus on reduced variance after being exposed to multi-modal stimuli reasonable. I think the authors make a great case for why in such variable systems this is a reasonable way to frame the interpretation of multimodal risk assessment.

Response: Thank you. We hope it will spur empirical tests!

I think an important source of variance is the time duration under which the response is measured: behavioral responses are immediate, but other physiological and life history responses are not. If I understand the results (in the supplement) correctly, physiological responses are less than behavioral responses (which I assume are the reference category) but life history and behavioral responses do not differ. I find this weird. Presumably the temporal order of responses is behavior (immediate), physiology (soon after), life history (longer after). Thoughts? What would happen if the same analyses were run separately on the different types of responses (behavioral, physiological, and life history)? There seem to be sufficient sample sizes...

Response: The referee understands the results presented in the supplementary results correctly; our meta-analysis reveals that behavioural response are strongest, followed by life-history responses, with physiological response showing the weakest response. However, we did not use any of the response categories as the reference. Instead, we present estimated effects for each response type such that any two responses types can be readily compared. In other words, the effect shown for physiological traits is not the difference relative to another category, but the mean effect size for physiological traits. Thus- to compare behavioural and physiological traits, one would look at the difference between the estimate for behavioural traits and the estimate for physiological traits.

We agree that the timing of response measurement is likely to matter, as is the duration of treatment. However, while our analysis shows no effect of timing at a coarse level (i.e., during treatment versus after treatment : <https://itchyshin.github.io/multimodality/#meta-regression-uni-moderator>; see “Response period” tab), there was an effect of treatment duration (in days) (<https://itchyshin.github.io/multimodality/#meta-regression-uni-moderator>; see “treatment duration” tab). Unfortunately, treatment duration was also confounded with trait type, making it difficult to parse out whether trait type of treatment duration was the main driver of differences between traits. We discuss this limitation here; lines 515-518.

Like the referee, we were also surprised that physiological responses were relatively weak. These included a range of traits including traits that are generally thought to be plastic on very short time scales (e.g., hormones), and traits that are plastic over longer time scales, such as days to weeks (e.g., body condition). Put another way- the relatively strong effect on life history was surprising, given that these traits are not generally considered to be plastic on as short of time scales as behaviour and physiology. However, this might be partly explained by the confound between response type and treatment duration; studies aimed at investigating life history responses to predation risk generally employed longer treatment durations (see Figures in online supplement: <https://itchyshin.github.io/multimodality/#meta-regression-multi-moderator>). This point is addressed in lines 509-518.

Although we have relatively good sample sizes for all trait types (Physiology: $k = 94$, Life history: $k = 112$, behaviour $k = 434$), we chose not to conduct separate analyses for each trait type for two reasons: 1), we did not have any a priori predictions that information integration would differ for these different trait types, and 2) the analyses would have been underpowered for V and AV cues within life-history and physiological traits. We haven't made any changes to the text with respect to this point. However, as more studies become available to synthesize, we agree this would be worth exploring.

Interestingly, there were no interactions with modality...

Response: Agreed! We were also surprised by this result, and particularly the lack of main effect of modality.

However, looking at the results, why are there both male and female estimates given I would assume one would be the reference category (as seen above with behavior being the reference category)? Is there a problem in your data set? Where am I going astray in interpreting these results?

Response: The results are presented correctly. We present estimated effect sizes for each level included (e.g., for analyses of sex: male, female both). This means that no specific level is the reference category against which the other two are compared. Instead, all pairwise contrasts are presented in a separate table. We've added detail to the methods and reworded the Figure legends to clarify.

Reviewer #1 (Remarks on code availability):

The code (which I did not run) looks well annotated.

Response: Thank you.

Reviewer #2 (Remarks to the Author):

The authors examined how predation cues presented alone or in combination can alter responses to predation risk in birds at many levels including behavioural and physiological. This topic has attracted much attention as responses to predation risk are fundamental in many taxa. This is a timely review of such effects and as far as I can tell the analysis was performed adequately.

Response: Thank you.

The results are a little underwhelming in the sense that overall the effect sizes were rarely different from 0 suggesting that researchers must rethink how they manipulate predation risk. This was not mentioned in the paper. The predictions regarding ranking of uncertainty according to cue type are rather simplistic, but this might not be the authors' fault as it is pervasive in the

literature. I agree with the authors that the current thinking, which focuses only on mean responses, needs refining. This being said, it was not clear what the paper was really proposing. The finding that variance in effect sizes varied depending on the combination of cues is interesting (although it was hard to judge from the figure) but why the authors looked at this was never explained. The idea was developed in the discussion making it difficult to really appreciate what it might mean. The paper works best when looking at the evidence based on the mean and would need an overhaul if the idea of variances is presented.

Response: The referee raises two points here, and we address each in turn.

The first point is that the overall effect sizes rarely differ from zero. This isn't accurate. There is strong support that birds respond to all types of manipulations of perceived predation risk (acoustic, visual, olfactory, combined cues), this is illustrated in figure 4. Specifically- the black rectangles represent the 95% CIs, and these are all markedly different from zero. We believe the referee may have mistakenly interpreted the whiskers in figures as 95% confidence intervals. The whiskers are 95% prediction intervals. This is defined in the figure legend, which we have reworded slightly to further clarify. The finding that all treatment types result in significant responses in focal birds suggests that manipulations of predation risk are indeed perceived as meaningful manipulations. However, in contrast with our expectation there is no evidence that the magnitude of response differs for different cue types. We rephrased the figure legends to clarify the distinction between 95% CI and 95% prediction intervals.

The second point that the referee raises is with respect to our finding that the amount of variance in response differs for unimodal versus multimodal cues. This result was not predicted *a priori*, but was discovered as a result of following the statistical guidance laid out in Nakagawa et al. (2015), which articulates why it is important to consider heterogeneous residual variances for such analyses. We have revised the methods to make the rationale for considering heterogenous residual variance clearer (lines 295-297 and line 299). We appreciate the referees point that the rationale for our proposal that integration of information from multimodal cues might lead to reduced variance is difficult to follow given that the idea was first introduced in the discussion. However, as this is a post-hoc interpretation of the results obtained in our meta-analysis, we feel strongly that it would be misleading to present this explanation as an *a priori* prediction in the introduction. We hope that the revisions we made to the methods (outlined above) have addressed the reviewers main concern with respect to this point.

Nakagawa, S., Poulin, R., Mengersen, K., Reinhold, K., Engqvist, L., Lagisz, M. *et al.* (2015). Meta-analysis of variation: ecological and evolutionary applications and beyond. *Methods Ecol. & Evol.*, 6, 143-152

It is a pity that the authors did not look at the evidence from other taxon beyond birds given that they are targeting a high-ranking journal with a broad audience. I have other minor comments listed below.

Response: Covering more taxa in our review would have had both advantages, and disadvantages. We opted to focus our review on avian taxa because we were already familiar enough with the literature to know that this would yield a large sample of studies

with relatively similar experimental designs to compare, providing us with ample statistical power (see lines 82-84). Although incorporating more taxa in our study would have provided greater breadth in some regards, it would have also introduced much more unexplained heterogeneity in the data set due to taxa-related differences in experimental designs and response variables. One advantage of having focused only on avian studies for this review is that future work synthesizing studies from other taxa can serve as independent tests of our hypothesis about multimodal cue integration. We highlight this opportunity in the discussion (lines 538-541).

Line 79: Are the results of a review and meta-analysis really a test of predictions? It seems to me that it is more an assessment of the evidence for or against predictions, not a test proper. Each included study is a test as it uses experimental manipulations of cues to examine responses to risk. The meta-analysis does not do that. Perhaps consider rephrasing this.

Response: Done. We have rephrased to state that the meta-analysis is being used to evaluate whether the existing body of empirical work support the predictions derived from existing theory (line 84).

Line 83: I am a little surprised to see olfactory cues in birds. I can certainly see how useful they can be for mammals but I thought that most birds were not really using such cues except birds like vultures. Can you elaborate on this or was this just an example?

Response: Indeed, until relatively recently, the potential role of olfaction was mostly overlooked in birds. However, there is now strong evidence that olfaction is an important sense in most birds. We have revised the text in the discussion section where we highlight the lack of studies using olfactory cues, and now provide a citation for the current consensus that birds do in fact have a strong sense of smell (lines 531-532).

Line 85: What about predator calls rather than calls by prey in response to a predator? Are such calls also considered less certain than visual cues? Please elaborate about predator calls, at I think such calls are used quite frequently in the literature. Contrary to a predator mount, for instance, a predator call suggests that a predator is present and also active. Not sure such calls would have less certainty than visual cues. Alarm calls by prey species are probably a little uncertain given that without visual cues they might be false alarms. Not knowing which calls were used in these experiments makes it difficult to be categorical about the rankings.

Response: The referee is correct that predator calls can also provide information about the presence/abundance of predators, and therefore potentially risk. We now list this as well (line 88). However, very few studies used predators calls in their acoustic manipulations of perceived predation risk. This is because predators generally do not vocalize when engaged in hunting. Thus, predator calls do not signal imminent risk of an attack, but rather, provide information that there are predators in the area that are not currently engaged in hunting, but which could potentially engage in hunting at a later time. Thus, when studies use predator calls, they typically do so to assess longer term responses (i.e., over the time scale where the likelihood that the predator will be hunting

is uncertain). Thus, acoustic cues, whether mobbing calls or predator calls, were both expected to provide lower certainty cues about predation risk. Because the vast majority of acoustic manipulations used mobbing/alarm calls, and because predator calls were only used for long-term manipulations of predation risk (e.g., over weeks to months), we could not meaningfully evaluate whether responses to mobbing calls differed from responses to predator calls. We have added this detail to the text to clarify (lines 94-98).

Line 99: Again the meta-analysis is not really making predictions but looking at the evidence in support of predictions from the primary literature.

Response: We agree with the referee that the meta-analysis is not making predictions. We have generated predictions based on the existing theoretical framework laid out in the introduction. We use the meta-analysis to test whether the body of empirical work are consistent with these predictions. We have made minor revisions in an attempt to clarify (lines 81, 84).

Line 111: So again only mobbing calls are considered? I would think that a visual cue along with a predator call would lead to cue enhancement in your framework. Please explain these choices.

Response: Thank you for pointing this out. Indeed, the only studies that used acoustic predator cues presented predator calls alone, not in combination, such that our only combined cues (acoustic + visual) include mobbing calls. We have now added this detail to the text (lines 427).

Line 129: Not all papers use birds or aves in the topic section. For instance, in a bird journal, it would be redundant (no pun intended) to add birds in the title, abstract, key words or topic. In your experience, was this not an issue?

Response: In total, 25 out of 116 studies (~20%) included in our review were from journals with a taxonomic focus on birds, thus, our search terms certainly did not exclude discovery of relevant literature from these sources. Our aim was not to provide a systematic map of relevant studies (i.e., find every single relevant study), but rather, to provide a systematic and reproducible means of obtaining an unbiased sample of studies relevant for addressing our questions. We have not made any changes to the text with respect to this point.

Line 184: What do you mean by associated datasets? I thought it was published papers that were reviewed. Please clarify.

Response: Many journals either require that complete data sets are published as a condition of manuscript acceptance, or authors chose to publish datasets to allow readers to evaluate the reproducibility of their results. In cases where the necessary summary statistics were not available from the information presented in the journal

article itself, but where the full dataset was available (e.g., in repositories such as Dryad, or Open Science Framework), we used the raw dataset to calculate the necessary summary statistic. In our case, all datasets were archived on Dryad. We have added detail to clarify this point (lines 198-199).

Line 187: Known to the authors. I can understand the logic of this but then again what if more were missed given my previous comment? I doubt that many have been missed. Given that you are the experts one can assume that this is probably a small proportion.

Response: For full transparency and reproducibility, we distinguished studies that were included simply because they were known to the authors, versus were captured using the specific criteria laid out in the methods. As these two studies represent only 16 out of 645 estimates used in our analysis (i.e. 2%), they do not alter the conclusions of our meta-analysis.

As articulated in response to a previous comment, the fact that some studies were undoubtedly missed in our systematic search criteria is not a problem so long as the studies we did capture reflect a random sample of available studies.

Line 191: I appreciate the effort to get as much information about each included study, but I was not always clear what was the motivation behind the collection of data for many variables (e.g. single vs multiple predators, stage of predation). All this information probably matters, but we were not told why it matters. This needs some justification.

Response: Thank you for pointing this out. Many of these variables were collected to allow us to explore whether they had a moderating effect on the response. Others were collected simply to provide more complete meta-data for readers. We now provide a table where we explicitly state the rationale for collecting each variable (Supplementary Information Table S4), which we refer to in this paragraph (lines 207-208 and 227-228).

Line 240: I was not clear about what observation id was. Can you elaborate?

Response: Observation id is a unique id value assigned to each observation (i.e., effect size estimate). We have rephrased to clarify (lines 264-265).

Line 242: Did you consider using a consensus tree instead of using one arbitrary tree?

Response: We did not use a single arbitrary tree, but rather used 50 posterior samples of the avian phylogenetic tree, which properly accounts for uncertainty in the phylogeny (see lines 266-270). We have not made any changes to the text in response to this point.

Line 248: This procedure seems to assume that all studies used the same subjects repeatedly. This is not necessarily the case. What does it mean for such studies when including this variance-covariance design?

Response: We apologise that our initial descriptions were incorrect. We have corrected the corresponding sentence to say “we created a variance-covariance matrix to add to meta-analytic models by assuming sampling variances from the same cohorts (subject ID) from the same studies have the correlation $r = 0.5$ ” (initially it only said “the same studies”). In our code, (subject_ID) has been used as a cluster to create the variance-covariance matrix.

Line 267: What is the prediction regarding the triple combination of cues? This was not presented in the introduction.

Response: We did not anticipate that there would be enough studies to allow for a meaningful test of the type of integration that occurs when all three cue types are presented together. Thus, we opted not to include the prediction and rationale in the introduction. Rather than provide a prediction retro-actively in the introduction, we have added text to the methods to acknowledge that we did not have a strong *a priori* prediction for the three integrated cues, nor the statistical power to address it given the lack of estimates ($k = 7$) and lack of studies ($k = 4$) (lines 295-297).

Line 288: For the less knowledgeable readers amongst us, can you elaborate on the idea of time lag. I am familiar with funnel plots but less so with other types of analyses. This journal is for a non-specialist audience.

Response: Time lag-bias occurs when larger or statistically significant effects are published earlier compared to small and/or statistically non-significant effects. We have now added this detail to the description of the time-lag bias analysis (lines 314-316).

Line 299: I am curious as to what type of olfactory cues were used in birds, but as I thought this was not frequent at all.

Response: The referee is correct that the use of olfactory cues remains relatively less common in birds compared to other taxa, as we highlight in our discussion (lines 532-533). When chemical cues of predators are used, they typically take one of two forms: 1) presenting material (e.g. paper towel or cotton swabs) that was previously in contact with a predator and has absorbed predator odor, or 2) presenting synthetic predator odours (i.e., chemical compounds that are present in predator anal gland secretions, feces, or urine). We have now added this detail to the text where we describe study inclusion criteria (lines 152-156).

Line 307: Judging from figure 4, none of the CIs exclude the value of 0. I was not clear what was the overall effect referred to here.

Response: We believe that the referee has misinterpreted the figures. We illustrate both 95% CIs (black rectangles) and 95% prediction intervals (whiskers). In fact, none of the CIs include the value of zero. These were already described in the figure legends, though we have revised the description slightly to clarify this point.

Line 332: I must admit that the lower variance is not at all obvious from this figure.

Response: We have added detail to the description of the results to clarify how the lower among-study variance can be inferred from Figure 4C (lines 362-364). Specifically, the 95% prediction interval is circa 50% smaller for the AV cue compared to either the A or V cue.

Line 340: Can you describe what a life-history response might be?

Response: Complete descriptions and examples of all categories of response variables (behavioural, physiological, and life-history) are provided in the Supplementary Information Table S2. We now refer to that table in the text here as well (lines 370-371).

Line 373: This idea that variance in responses might vary depending on the treatment combination was not explained in the introduction. As presented here, it almost appears as a post-hoc finding.

Response: The referee is correct that we had not explicitly predicted reduced variance in response as an outcome of multi-modal cue integration, which is why the explanation is presented post hoc in the discussion, rather than as an a priori prediction in the introduction. Consideration of heterogeneous variances was done because data visualizations suggested that variances differed across treatment groups, and it is recommended that meta-analysis explicitly consider treatment effects on both means and variances as standard practice, even when effects on variances are not predicted. We have revised the text to clarify this point (lines 295-299).

Line 385: To return to my previous point, it would seem that a predator call might be just as relevant as a visual cue alone.

Response: This point has already been addressed in response to the referee's earlier point (see above).

Line 392: Can we really have strong support for no effect? The evidence to me suggests that the evidence for a difference is weak.

Response: Yes, it is possible to have strong support for no effect; when the estimated effect size is close to zero, and the 95% CI is narrow, this is strong support for no effect. This is different from weak support for an effect (which would be an estimated effect that

is biased away from zero, but whose confidence intervals overlap zero), or strong support for a weak effect (which would be an estimated effect that is non-zero but small, and whose 95% CI does not overlap zero). We have not made any changes to the text with respect to this point.

Line 395: To reiterate my point, perhaps it would be helpful to use type of acoustic cue used as a moderator variable or at least to present a breakdown of the different types of acoustic cues used.

Response: We appreciate the referees point that the type of acoustic cue could matter. However, as articulated in our response to their earlier point, in practice, almost all acoustic cues used in the studies synthesized here were mobbing and/or alarm calls by the focal species. Given the very small number of estimates and studies that used predator calls, it would not be possible to include type of acoustic cue as a moderator. We have not made any additional changes to the text beyond those made in response to the referee's earlier point.

Line 412: This topic of MLE integration should have been introduced earlier as it seems very relevant to the ideas tested here. I got the impression that this was added following the results. Given the complexity of the ideas expressed here, more than a cursory treatment is needed.

Response: The referee is correct that the topic of MLE integration was introduced only following the results. This is because it is our *post hoc* interpretation of the observation that among-study variance decreases when multimodal cues are presented relative to unimodal cues. While we can appreciate that the discussion of MLE integration might be easier for readers to follow if we primed them for it first in the introduction, it would be misleading to present as part of our *a priori* framework, and we have therefore not made changes to the introduction with respect to this point. However, we have carefully revised the discussion and text box to improve clarity and we hope that this has addressed the referee's concern.

Line 506: Passerines are very numerous. Can this alone explain why they appear more commonly in such studies?

Response: The referee is correct that passerines are the most species rich group of birds (~6500 species), representing approximately 2/3 of all bird species. However, passerines represented 542 out of 645 estimates in our meta-analysis (84% of all estimates), and are thus over-represented despite comprising the majority of bird species. We have now added this detail to the text to clarify (lines 544-545).

Reviewer #3 (Remarks to the Author):

In this paper, the authors perform a meta-analysis to test whether and how uni-modal versus (simultaneous) multi-modal cues of potential predation risk affect anti-predator responses

among birds. I was actually quite impressed with the number of studies that have performed this type of experiment and so I think this meta-analysis does a really nice job of summarizing this literature and providing some concrete conclusions and considerations for future work. I would now actually be quite excited to see a similar meta-analysis done on another taxonomic group (my guess would be fish would be the next biggest set of studies) as I could imagine that we might expect the cue responses to be similar in some regards (visual cues probably offer highest certainty) and different in others (olfactory cues might offer far more information in aquatic systems, especially given that many studies have manipulated both predator chemical cues and alarm cues from injured prey). But anyway, that's just an idea for future work and I only mention it because it speaks to how interesting I found this paper and that it spurred my own thoughts on future work quite a bit!

Response: Thank you for your comments. We agree that it would be timely to follow up this meta-analysis with work in other taxa where similar types of experiments (i.e., unimodal and multimodal manipulations of predation risk, mate quality, etc.) have also been conducted to provide an independent test of the MLE hypothesis we propose in the discussion.

But altogether, my impression of this paper is very positive. It's well-written and easy to follow. The authors appear to have followed all the most up-to-date guidelines in how to perform systematic and reproducible meta-analyses (e.g. explicit flowcharts showing inclusion/exclusions criteria; PRISMA checklist; accessible code etc). As someone who's also done several meta-analyses themselves, this all looks completely in order and a great example of clear reporting of their methods.

Response: Thank you.

So only real comments are in some areas where I think a bit more detail or expansion might be useful.

1 – I wonder if it might be useful to add another paragraph or so to the introduction that really expands on how cues might differ in the expected change in the magnitude of the response based on the intensity of the cue, and the certainty of the cue. Both of these topics are brought up in the first paragraph of the intro together. I'm pretty well steeped in the cue certainty literature myself so I quickly caught the difference between these aspects of cue information, but a more naïve reader might benefit from a bit more detail about how these axes differ from each other and how they combine.

Response: Thank you for this suggestion. We have expanded the detail in what was previously the first paragraph of the introduction. It is now split in two paragraphs, with the first paragraph highlighting effects of cue magnitude, and the second cue certainty (see lines 40, 44-47, and 48-49).

2 – In the results on Line 306, the authors state that “birds responded in the predicted direction” which then made me realize that I don't think I saw any mention in the methods about exactly

how these directions were coded from the data extraction? First off, what is the predicted direction (increase or decrease)? And who predicted this – did the authors make these predictions and so code the responses themselves, or did they follow whatever the authors of the included studies predicted? I think either way is likely fine, but it should be mentioned. Would be useful to have some more detail here about how to interpret these: which types of behaviors/physiological responses were expected increase versus decrease in response to predation threat? I don't think this affects the interpretation at all, but some more biological detail would be nice.

Response: Thanks for pointing this out. Supplementary Information Table S2, which is referenced immediately following this statement, provides a detailed list of all response variables extracted, and the predicted effect of increased perceived predation risk on the mean response. We have now added some text to make this more explicit (lines 240-246).

3 – I find the mention of maximum likelihood estimation as an interpretation of the variance results very interesting. Reading through this immediately made me think though about why would we expect to see lower variance at the among-SPECIES level if MLE is clearly being done at the individual level. But I was then happy to see that authors anticipated this and had a clear discussion of how these two levels may interact and so completely answered my question. My guess is that this result (that audio plus visual cues results in lower study variance) will actually be very informative for future work as it clearly demonstrates that presenting both cues together is more likely to elicit the appropriate response from the species than either presented alone. My guess is this paper should be heavily cited as a justification for this method in the future.

Response: Thank you! We're happy that our discussion of how MLE could apply across different biological levels was clear, and we hope it will be a useful framework for future work.

Reviewer #3 (Remarks on code availability):

I checked the github page they cite in their paper, which nicely presents all of their results (including code and figures). I didn't run the code on my own computer, but quickly read through it online and it looks well documented and the figures match the paper which is always a good sign.

Response: Thank you for cross checking our data/code repository against results presented in the main text.

REVIEWERS' COMMENTS

Reviewer #1 (Remarks to the Author):

I'm satisfied with this revised version.
Dan Blumstein

Reviewer #1 (Remarks on code availability):

Not this version--I reviewed the original code.

Reviewer #2 (Remarks to the Author):

Thank you for considering my comments. Job well done. I have no further comments.